# Practice of Awake Prone Positioning in Critically Ill COVID-19 Patients—Insights from the PRoAcT–COVID Study

**DOI:** 10.3390/jcm11236988

**Published:** 2022-11-26

**Authors:** Willemke Stilma, Christel M. A. Valk, David M. P. van Meenen, Luis Morales, Daantje Remmelzwaal, Sheila N. Myatra, Antonio Artigas, Ary Serpa Neto, Frederique Paulus, Marcus J. Schultz

**Affiliations:** 1Department of Intensive Care, Amsterdam University Medical Centers, Location ‘AMC’, 1105 AZ Amsterdam, The Netherlands; 2Center of Expertise Urban Vitality, Faculty of Health, Amsterdam University of Applied Sciences, 1105 BD Amsterdam, The Netherlands; 3Department of Anesthesiology, Amsterdam University Medical Centers, Location ‘AMC’, 1105 AZ Amsterdam, The Netherlands; 4Servei de Medicina Intensiva, Hospital Universitari Sant Pau, 08025 Barcelona, Spain; 5Translational Research Laboratory, Institut d’Investigació i Innovació Parc Taulí I3PT, Universitat Autònoma de Barcelona Sabadell, 08208 Barcelona, Spain; 6Department of Anaesthesiology, Critical Care and Pain, Tata Memorial Hospital, Homi Bhabha National Institute, Mumbai 400012, India; 7Intensive Care Department, CIBER Enfermedades Respiratorias, Parc Tauli University Hospital, 08208 Sabadell, Spain; 8Autonomous University of Barcelona, 08193 Sabadell, Spain; 9Australian and New Zealand Intensive Care, Research Center (ANZIC-RC), Monash University, Melbourne 3800, Australia; 10Department of Critical Care Medicine, Hospital Israelita Albert Einstein, São Paulo 05652-900, Brazil; 11Centre for Tropical Medicine and Global Health, Nuffield Department of Medicine, University of Oxford, Oxford OX3 7LG, UK; 12Mahidol–Oxford Tropical Medicine Research Unit (MORU), Faculty of Tropical Medicine, Mahidol University, Bangkok 10400, Thailand

**Keywords:** coronavirus disease 2019, COVID-19, acute hypoxemic respiratory failure, awake prone positioning, self-proning, prone positioning, outcome, propensity matching

## Abstract

We describe the incidence, practice and associations with outcomes of awake prone positioning in patients with acute hypoxemic respiratory failure due to coronavirus disease 2019 (COVID-19) in a national multicenter observational cohort study performed in 16 intensive care units in the Netherlands (PRoAcT–COVID-study). Patients were categorized in two groups, based on received treatment of awake prone positioning. The primary endpoint was practice of prone positioning. Secondary endpoint was ‘treatment failure’, a composite of intubation for invasive ventilation and death before day 28. We used propensity matching to control for observed confounding factors. In 546 patients, awake prone positioning was used in 88 (16.1%) patients. Prone positioning started within median 1 (0 to 2) days after ICU admission, sessions summed up to median 12.0 (8.4–14.5) hours for median 1.0 day. In the unmatched analysis (HR, 1.80 (1.41–2.31); *p* < 0.001), but not in the matched analysis (HR, 1.17 (0.87–1.59); *p* = 0.30), treatment failure occurred more often in patients that received prone positioning. The findings of this study are that awake prone positioning was used in one in six COVID-19 patients. Prone positioning started early, and sessions lasted long but were often discontinued because of need for intubation.

## 1. Introduction

Patients with severe coronavirus disease 2019 (COVID-19) can develop profound hypoxemia that is refractory to low-flow and even high-flow oxygen supplementation [1]. Prone position has been shown to improve outcomes in ventilated ARDS patients [2]. Several changes, induced by prone positioning, could be responsible for this benefit [3]. First, in a prone position the heart is no longer causing atelectasis because of the changed position in the thorax. Additionally, the chest wall compliance changes and there is a more even distribution of aeration when turning a patient prone. The latter is associated with a better ventilation–perfusion ratio. This all allows changes in ventilator setting, specifically those that are associated with ventilator-induced lung injury. Last but not least, prone position could reduce the afterload of the right ventricle, and with prone position the direction of the trachea is downward, which could help sputum evacuation.

The use of prone positioning in non-intubated patients, so called awake prone positioning, was initially reported in case series to improve oxygenation in patients with COVID-19 [4]. Additionally, cohort studies showed the potential to improve outcomes in patients with acute hypoxemic respiratory failure due to COVID-19 [5,6]. One randomized clinical trial reported an improved outcome with the use of awake prone positioning in patients with respiratory failure due to COVID-19 [7]. Another randomized clinical trial did show a lower intubation rate, but failed to show a difference in mortality [8].

Based on available literature and clinical experience, practical guidelines for prone positioning of non-intubated patients became rapidly available after the initial wave of the pandemic [9,10,11,12]. Since then, the use of awake prone positioning has increased. However, results on practical aspects of awake prone positioning in patients after admission to the intensive care unit (ICU), as well as its relation with clinical outcomes, are relatively scarce [5,6,7].

How frequent awake prone positioning was used in the ICU during the second wave of the national outbreak in the Netherlands, which occurred after the change in the guidelines, is unknown. It is also uncertain how it was applied and whether it was associated with patient characteristics and outcome. Therefore, we analyzed the database of a large observational study, named ‘Practice of Adjunctive Treatments in ICU Patients with COVID-19’ (PRoAcT–COVID) [13].

The primary aim was to determine the practice of prone positioning in patients that did not immediately proceed with invasive ventilation after arrival in the ICU. The second aim was to compare epidemiology and outcomes in patients that received prone positioning versus patients that received standard care. We hypothesized that prone positioning was used often, and that its use had associations with outcome.

## 2. Materials and Methods

### 2.1. Study Design

PRoAcT–COVID is an observational cohort study in critically ill patients with COVID-19 that were admitted to ICUs of the participating hospitals in the Netherlands. In PRoAcT–COVID, we captured data in the first 3 months of the second wave of the national outbreak regarding pharmacological and non-pharmacological interventions. The protocol was approved by the local Institutional Review Board of the Amsterdam University Medical Centers, location ‘AMC’ (W20_526 # 20.583), and thereafter in all participating centers. Need for individual patient informed consent was waived seen the observational nature of the study. The study plan and the analysis plan for this current analysis were both pre–published [13,14]. PRoAcT–COVID is registered at clinicaltrials.gov with trial number NCT04719182. Registered 22 January 2021.

### 2.2. Patients

We screened patients that were admitted to the ICUs between October 2020 and January 2021 on admission day. Inclusion criterium was an admission to the ICU for acute hypoxemic respiratory failure due to RT–PCR confirmed COVID-19. Acute respiratory failure was defined as the need for hypoxemia refractory to oxygen supplementation with low flow oxygen systems, such as nasal prong or simple face masks. Patients aged < 18 years and patients with an alternative diagnosis were excluded.

For the current preplanned descriptive analysis of the PRoAcT–COVID study we excluded patients that were under invasive ventilation at ICU admission or intubated immediately; immediate intubation was defined as intubation that happened within the first 2 h after arrival in the ICU. We also excluded patients that were transferred from or to ICUs from other non-participating hospitals within the first 2 calendar days of ICU admission. This was carried out because of two reasons: first, we were not allowed to capture data of patients in the non-participating centers; second, we could not exclude the possibility that an imminent transport to another ICU may have influenced decisions to intubate the patient. At the time of this study, scheduling was chaotic, often not knowing exactly when a patient would be transported and patients frequently had to be moved within one or two hours. Since the policy was to transfer a patient only when intubated, this meant that patients were frequently intubated while awaiting transport.

### 2.3. Collected Data

We collected baseline and demographic variables, including sex, age, weight and height, major comorbidities, and home medication, the first day with symptoms, the day of the definite diagnosis of COVID-19, the day of hospital and ICU admission. The day of ICU admission, which in theory could last from 1 min to 23 h and 59 min, was named ‘day 0’. Successive days were named ‘day 1’, ‘day 2’ onwards to ‘day 28′. The Simplified Acute Physiology Score (SAPS) II was calculated at 24 h after ICU admission [15] and extracted from routinely captures and in part automatically generated data in the patient data management systems as present in the participating ICU.

We collected data regarding prone positioning, including time of initiation, the number of sessions per day, and the number of days it was continued. The data collectors were trained to collect data on prone positioning from the data patient management system, wherein body position is always reported. This was further confirmed by reading each nursing report. This allowed us to determine whether a patient ever was placed in a prone position, and also the timing of the intervention. We also collected detailed information regarding oxygen support until 28 days or ICU discharge, including the type of oxygen interface, i.e., nasal sprong or cannula, non-rebreather mask or Venturi mask, continuous positive airway pressure (CPAP), high–flow nasal oxygen (HFNO), and noninvasive and invasive ventilation.

Up to day 28, we captured intubation status, mortality and date of ICU and hospital discharge.

### 2.4. Patient Classification

To describe current practice of awake prone positioning we classified patients into two groups. The group of patients that were placed in the prone position, hereafter named the ‘prone positioning group’, were compared to the group of patients that were not placed in prone position, hereafter named the ‘standard care group’.

### 2.5. Endpoints

The primary endpoint was the practice of awake prone positioning, including the following aspects: timing, duration of prone sessions, frequency, and for how long prone positioning sessions were repeated. One secondary endpoint was ‘treatment failure’, a composite endpoint of intubation or death before day 28—this endpoint was used in a meta-trial of awake prone positioning in patient with acute hypoxemic respiratory failure due to COVID-19 [7]. We also report the two components separately, as well as ICU and hospital length of stay, and ICU and hospital mortality.

### 2.6. Power Calculation

Since this is a preplanned descriptive analysis of an observational cohort study, we performed no a priori sample size calculation; the number of patients admitted in the participating centers served as the convenience sample.

### 2.7. Statistical Analyses

Categorical variables are presented as counts (frequencies), and continuous variables are presented as medians (interquartile ranges [IQRs]). Independent categorical variables were compared with Fisher exact test, and continuous variables with Wilcoxon rank sum test.

The incidence of awake prone positioning is expressed as the proportion of patients that was admitted to the ICU not yet intubated and not intubated early after ICU admission. Timing of prone positioning is expressed as the number of days between ICU admission and start of the first session of awake prone positioning. Duration of prone positioning is expressed as the mean number of hours of each session. Frequency is expressed by the number of sessions per days a patient received awake prone positioning. Continuation of prone positioning is expressed as the number of days a patient received one or more awake prone positioning sessions. Types of oxygen support used at the day of initiation of awake prone positioning and thereafter are reported for the awake prone positioning group. The endpoint treatment failure, and its two components, are visualized using Kaplan–Meier curves, as were the endpoints length of stay in ICU and hospital.

Since the exposure was not randomly assigned, we also performed a propensity matched analysis to control for known confounders. This approach was chosen to account for the fact that the exposure to the intervention of interest might not occur during the study if improvement occurs or termination of efforts, e.g., because of intubation or death, occurred first. The propensity score was calculated by means of a generalized linear model based on a (shared-frailty) Cox proportional hazards model with exposure during follow–up as the dependent variable and based on baseline characteristics and daily information. The following baseline variables were included based on clinical relevance: age; gender; and body mass index (BMI). In addition, the following covariate assessed daily was included: PaO_2_. The propensity score for each patient was then derived from the Cox model as the hazard component (i.e., the linear predictor) at any given moment from the model. A 1:3 risk set matching on the propensity score was performed using a nearest neighbor–matching algorithm with a maximum caliper of 0.01 of the propensity score. A strict margin of the maximum caliper of the propensity score was chosen to match patients. With this, some patients may not be matchable, resulting in a lower number of patients in the matched analysis. Patients receiving the intervention at any given moment were separately and sequentially propensity score matched with a patient who was at risk of receiving the intervention within the same moment. At-risk patients included those who were still undergoing treatment and did not receive the intervention before or within the same moment. At-risk patients therefore also included patients who received the intervention later, as the matching should not be dependent on future events [16]. As such, the matched group with no intervention includes patients who subsequently received the treatment (although later than their matched counterpart). In all analyses, the time-dependent exposure was considered a stochastic process (counting process) that equals zero from time 0 until exposure, then it equals to one until the end of observation. This provided a correction for the possibility of immortal time bias. The performance of matching was assessed through standardized differences between baseline characteristics.

Binary outcomes will be compared with mixed-effect generalized linear models with binomial distribution and expressed as odds ratio and 95% confidence interval (CI). Continuous outcomes will be compared with mixed-effect generalized linear models with Gaussian distribution and expressed as mean difference and 95% CI. Time-to-event outcomes were assessed with shared–frailty Cox proportional hazard models. ICU length of stay will be analyzed with a clustered Fine–Gray competing risk model with death before the event as competing risk. In all models the hospitals will be included as random effect to account for potential clustering. Wherever appropriate, Kaplan–Meier curves are used to present time-to-event outcomes. All tests were two-sided, with a significance level set at <0.05. All statistical analyses were conducted using R v.4.0.3 (R Foundation for Statistical Computing: Vienna, Austria).

## 3. Results

### 3.1. Patients Enrolled

Sixteen ICUs from various types of hospitals, including academic, teaching and non-teaching hospitals participated in PRoAcT–COVID, and a total of 946 patients were enrolled. Of these, 546 patients were eligible for this preplanned descriptive analysis. Main reason for exclusion was admission under invasive ventilation or immediate intubation after ICU admission (Figure 1).

Most patients were male, and were overweight or obese. Patients were severely ill, as reflected by high disease severity scores, and had severe hypoxemia despite high levels of FiO_2_ that was often supplied by means of HFNO. After hospital admission, patients were admitted to the ICU within one or two days (Table 1).

### 3.2. Practice of Awake Prone Positioning

Awake prone positioning was used in 88 (16.1%) patients. Patients were placed in a prone position median 1 [0 to 2] days after ICU admission. Total time of awake prone positioning summed up to median 12.0 [8.4–14.5] hours for median 1.0 day.

HFNO was the most often used oxygen interface during awake prone positioning (79.5%), followed by CPAP (9.1%) (Table 2).

### 3.3. Epidemiology

There was no difference in the incidence of do-not-intubate orders (Table 1). Patients in the prone positioning group less often had a history of cardiovascular disease (Table 1). Other baseline characteristics were not different between the two groups. Patients in the prone positioning group had worse oxygenation, but oxygen supplementation was comparable to that in the standard care group (Table 3).

### 3.4. Treatment Failure

81 (92.0%) patients in the prone positioning group versus 289 (63.1%) patients in the standard care group experienced treatment failure, the primary endpoint of this analysis (Figure 2). The difference in treatment failure was driven by a difference in the intubation rate before day 28, and not by a difference in mortality.

### 3.5. Other Outcomes

ICU and hospital length of stay were significantly longer in the prone group than in the standard care group (Appendix A); ICU and hospital mortality were not different between the groups (Appendix A).

### 3.6. Propensity Matched Analysis

We matched 70 patients in the prone positioning group to 175 patients in the standard care group (Appendix A). Of 88 patients that received awake prone position, 18 patients could not be matched due to the strict margin of maximum caliper of the propensity score. Differences in treatment failure, mortality, and intubation rate before day 28 remained but were not statistically significant (Figure 2, Appendix A).

## 4. Discussion

The findings of this preplanned descriptive analysis of the national multicenter observational cohort PRoAcT–COVID can be summarized as follows: (1) awake prone positioning was used in one in six patients with refractory hypoxemia due to COVID-19; (2) prone positioning started early after ICU admission, and sessions lasted for many hours but it was often stopped early because of need for intubation; (3) the only patient characteristic that had an association with prone positioning was a history of cardiovascular disease, which was more often present in standard care patients; and (4) in unmatched analysis, but not in matched analysis prone positioning had an association with treatment failure, an association that was mainly driven by a higher intubation rate.

Our study has several strengths. PRoAcT–COVID included a large proportion of all severely ill acute hypoxemic COVID-19 patients that were admitted to ICUs during the second wave of the pandemic in the Netherlands. We focused on patients that were admitted to the ICU but did not start immediately with invasive ventilation at ICU admission—in other words, patients that were severely ill and could have received the intervention of interest. We included various types of hospitals, including academic, teaching and non-teaching hospitals, contributing to the generalizability of the findings. Follow-up was complete, and there were no missing data for the primary analysis. Finally, we strictly adhered to the preplanned statistical analysis plan, and used sophisticated analyses including propensity matching to reduce the risk of confounders.

The findings of our study expand our knowledge on the practice of prone positioning in non-intubated acute hypoxemic COVID-19 patients. One RCT in intubated patients with a low P/F ratio showed that use of prone positioning sessions for at least 16 h to be effective [2]. Recently a systematic review reported that treatment of patients with COVID-19 was not so different compared to patients with classic ARDS [17]. Additionally, one recent meta-analysis showed that early use of prone positioning in non-intubated patients with acute hypoxemic respiratory failure due to COVID-19 could be beneficial [18]. However, use of awake prone positioning and its continuation seems quite variable, and details are not always clearly reported [18]. The incidence of prone positioning in our cohort was lower compared to that in one other study [6], but similar to that in another study [19]. Duration of sessions were comparable to that in three studies [5,19,20], but longer than that in another report [12]. Of interest, sessions lasted much longer than those in studies performed in general wards [18].

Our study shows that prone positioning was used for one day in the majority of patients. This is relatively short compared to other studies, reporting that prone positioning was used for two days or longer [18]. This could affect, at least in part, the outcomes in our cohort. Differences with regard to duration of use of prone positioning between studies could be explained by differences in settings—we studied patients that were admitted to the ICU, patients that were probably much sicker but whose care was provided by more experienced healthcare providers than in the general ward [18]. Another explanation could be that in the Netherlands, the use of awake prone positioning was relatively new at the moment of data collection and an expert consensus statement about management strategies advising awake prone positioning, was published afterwards [9].

Our findings are supported by the results of a multicenter retrospective cohort study in 501 patients with COVID-19 and hypoxemia [21]. In that study, prone positioning offered no clinical benefit in patients who had not received mechanical ventilation. In addition, the odds of having a worse outcome at day 5 based on a modified World Health Organization ordinal scale was higher among patients receiving the awake prone positioning intervention, suggesting potential harm of this intervention in patients with acute hypoxemia due to COVID-19. Taken together, routine recommendation for awake prone positioning may not be beneficial in these patients.

The high proportion of patients receiving HFNO during awake prone positioning, is comparable to that reported in other reports [5,6,22]. Other oxygen interfaces were seldom, or not, used in our cohort. It could be that CPAP, non-invasive ventilation and ventilatory support with the helmet is relatively underused in the Netherlands, at least in part explaining the low incidences of these forms of oxygen support in this cohort.

The one-single difference in baseline characteristics between patients that received awake prone positioning and patients that did not receive this intervention, was a history of cardiovascular disease. Although it has been reported before that prone positioning could improve cardiac function [3], placing a patient in a prone position could also lead to hemodynamic instability, e.g., because of increased need for sedation—this may have led to less enthusiasm to use prone positioning in those patients [2]. This, however, remains speculative as we were not able to collect data on sedation practice.

Treatment failure occurred more often in the awake prone positioning group than in the standard care group. This finding may seem in contrast with findings of the abovementioned meta-trial of awake prone positioning [7]. Indeed, in that meta-trial treatment failure occurred less often in patients that received awake prone positioning compared to patients that received standard care. It should be realized, though, that studies in this meta-trial compared awake prone positioning with standard care in COVID-19 patients within the hospital, i.e., in patients in a general ward or in an ICU [7]. Substantial heterogeneity was also shown in a recent meta-analysis of observational studies, reporting that use of awake prone positioning was associated with a reduced mortality but not with a lower intubation rate [20]. Awake prone positioning may translate in a better outcome more if it is applied early, i.e., as a preventive measure in a general ward in patients receiving high flow oxygen support, and less if it is applied late, as a rescue treatment in an ICU. Of note, a comparable difference in treatment effects in relation to the moment of initiation was recently shown regarding therapeutic anticoagulation with heparin in COVID 19 patients [23,24].

The association of awake prone positioning in the unmatched-cohort with treatment failure was driven by a difference in need for intubation. One could hypothesize that the association with treatment failure results from a causal relation. Awake prone positioning could increase sputum evacuation, possibly creating an acute need for intubation in case massive amounts of sputum enter the larger airways. It should be noticed, though, that in matched analysis there were no associations of prone positioning with treatment failure.

Awake prone positioning may have the potential to improve outcomes of patients with acute hypoxemic respiratory failure due to another cause. On the other hand, awake prone positioning is an intensive and time-consuming intervention that may require training of healthcare professionals for save application. Therefore, randomized evidence is needed. One recently started multicenter randomized clinical trial investigates the effect of awake prone positioning on intubation rate and mortality in ICU patients with acute hypoxemic respiratory failure not necessarily caused by COVID-19 [25].

While a multivariate analysis may have been preferred to identify patient characteristics that are independently associated with the use of awake prone positioning, data on specific factors such as respiratory rate and the work of breathing could not be collected, and the sample size would be too small to draw any meaningful conclusions.

PRoAcT–COVID has limitations. As this was an observational study, variety in practice of care and reporting could have caused inaccuracies in the data. Despite the presence of a new national guideline for use of awake prone positioning, the decision to apply awake prone positioning could still have been based on clinical expertise and reasoning by the ICU team. In line with the study design of the PRoAcT–COVID study we could not collect data on practice of awake prone positioning before admission to the ICU, complications related to awake prone positioning, and changes in oxygenation or oxygen supply during awake prone positioning. While there has been evidence for involvement of the cardiovascular system in COVID-19 and COVID-19 related outcomes, unfortunately we did not collect data regarding cardiovascular complications. Selection of ICUs for this study, which was based on previous collaborations in studies of ventilation in critically ill patients, including those with COVID-19 patients [26], may have resulted in an over–representation of ICUs with more knowledge and experience in awake prone positioning. In addition, the exclusion criteria used for this preplanned descriptive analysis may have resulted in selection bias. Finally, due to the observational nature of our data, no causal relationship can be established and the findings of the descriptive analysis in this study should be regarded as exploratory.

## 5. Conclusions

In this national multicenter observational cohort, awake prone positioning was used in one in six critically ill acute hypoxemic COVID-19 patients. In the unmatched analysis patients that received prone positioning had higher risk for treatment failure. However, this was not confirmed in the matched analysis. We are in urgent need for randomized clinical trials of prone positioning in non-intubated patients with acute hypoxemia.

## Figures and Tables

**Figure 1 jcm-11-06988-f001:**
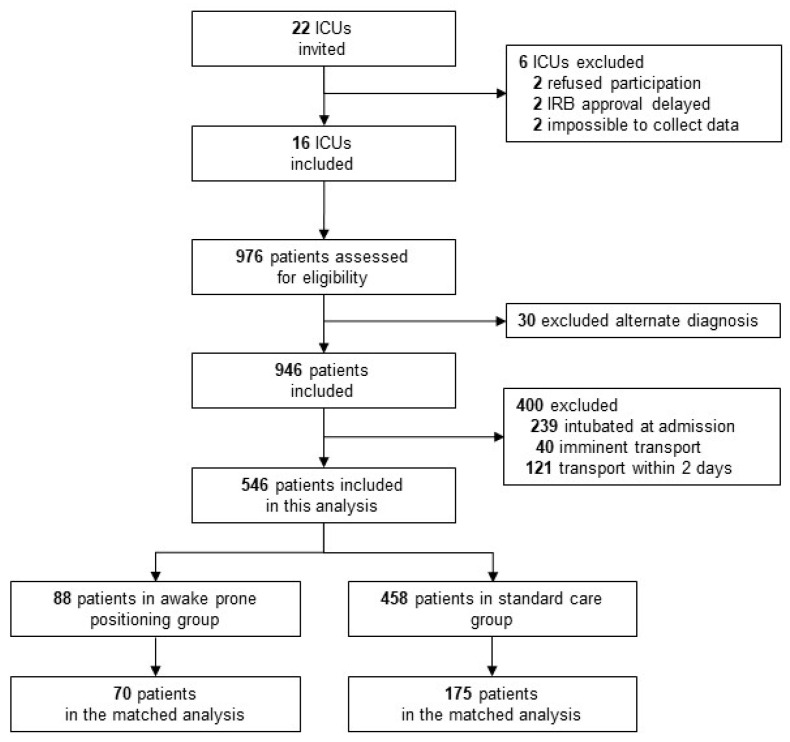
Flow chart of patient inclusion.

**Figure 2 jcm-11-06988-f002:**
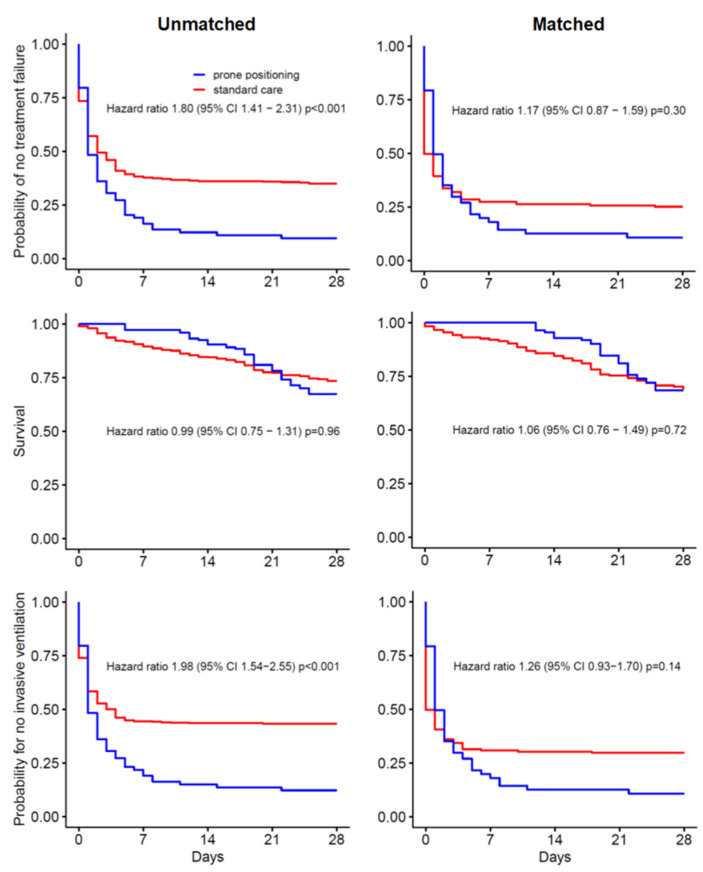
Patient outcomes in the unmatched (**left** panel) and the matched analysis (**right** panel).

**Table 1 jcm-11-06988-t001:** Baseline characteristics.

	Overall *N* = 546	Prone Positioning *N* = 88	Standard Care *N* = 458	*p*
Age, years (median, IQR)	67 (59–73)	66 (60–73)	67 (59–73)	0.946
Male gender, N (%)	402 (73.6)	60 (68.2)	342 (74.7)	0.257
BMI, kg/m^2^ (median, IQR)	28.0 (25.3–31.8)	28.8 (25.7–31.9)	27.9 (25.2–31.8)	0.543
SAPS II score (median, IQR)	43 (33–53)	47 (37–53)	43 (32–53)	0.056
Do-not-intubate order, N (%)	31 (5.7)	2 (2.3)	29 (6.3)	0.209
Comorbidities				
Arterial hypertension, N (%)	219 (40.1)	39 (44.3)	180 (39.3)	0.447
Cardiovascular disease, N (%)	135 (24.7)	14 (15.9)	121 (26.4)	0.050
Heart failure, N (%)	30 (5.5)	2 (2.3)	28 (6.1)	0.233
COPD or asthma, N (%)	97 (17.8)	16 (18.2)	81 (17.7)	1.000
Diabetes mellitus, N (%)	178 (32.6)	24 (27.3)	154 (33.6)	0.298
Chronic kidney disease, N (%)	50 (9.2)	7 (8.0)	43 (9.4)	0.822
Malignancy, N (%)	39 (7.1)	6 (6.8)	33 (7.2)	1.000
Neuromuscular disease, N (%)	13 (2.4)	2 (2.3)	11 (2.4)	1.000
Obstructive sleep apnea, N (%)	39 (7.1)	6 (6.8)	33 (7.2)	1.000
Days in hospital before ICU admission, (median, IQR)	1.0 (0.0–4.0)	2.0 9 (0.0–3.0)	1.0 (0.0–4.0)	0.085

*Abbreviations: BMI: Body Mass Index; SAPS: Simplified Acute Physiology Score; COPD: Chronic obstructive pulmonary disease and/or asthma.*

**Table 2 jcm-11-06988-t002:** Oxygen supplementation at start of prone positioning.

	Prone Positioning *N* = 88
HFNO, N (%)	70 (79.5)
FiO_2_, % (median, IQR)	82 (65–95)
Air flow, L/min (median, IQR)	60 (50–60)
CPAP, N (%)	8 (9.1)
FiO_2_, % (median, IQR)	67.5 (63.8–93.3)
Non-Rebreather or Venturi Mask, N (%)	5 (5.7)
Oxygen, L (median, IQR)	15.0 (15.0–15.0)
NIV, N (%)	4 (4.5)
PEEP, cmH_2_O (median, IQR)	8.0 (7.3–9.3)
FiO_2_, % (median, IQR)	72.5 (57.5–86.3)
SpO_2_, % (median, IQR)	91.3 (89.0–94.0)
PaO_2_, mmHg (median, IQR)	72.0 (60.0–84.5)

*Abbreviations: CPAP: Continuous Positive Airway Pressure, HFNO: High Flow Nasal Oxygen, NIV: Non-invasive ventilation.*

**Table 3 jcm-11-06988-t003:** Oxygen supplementation and characteristics at ICU admission day.

	Overall *N* = 478	Prone Positioning *N* = 88	Standard Care * *N* = 390	*p*-Value
Oxygen support **				
Not known	3 (0.6)	0 (0.0)	3 (0.8)	
Nasal sprong, N (%)	24 (5.0)	1 (1.1)	23 (5.9)	
Oxygen, L (median, IQR)	4 (3–5)	5 (5–5)	4 (3–5)	0.376
Non-Rebreather or Venturi Mask, N (%)	60 (12.6)	9 (10.2)	51 (13.1)	
Oxygen, L (median, IQR)	15 (12–15)	15 (15–15)	15 (12–15)	0.038
High Flow Nasal Oxygen (HFNO), N (%)	372 (77.8)	73 (83.0)	299 (76.7)	
FiO_2_, % (median, IQR)	80 (60–90)	80 (60–94)	75 (60–90)	0.161
Flow, Liters oxygen/min (median, IQR)	50 (50–60)	50 (50–60)	50 (50–60)	0.057
Non-invasive ventilation (NIV), N (%)	13 (2.7)	2 (2.3)	11 (2.8)	
PEEP, cmH_2_O (median, IQR)	6 (5–8)	9 (7–11)	6 (5–8)	0.688
FiO_2_, % (median, IQR)	50 (40–60)	55 (52.5–57.5)	50 (40–65)	0.481
Missing data, N (%)	6 (0.01)	3 (3.4)	3 (0.7)	
Respiratory values ***				
SpO_2_, % (median, IQR)	93 (90–95)	91 (89–94)	93 (90–96)	<0.001
PaO_2_, mmHg (median, IQR)	76 (25–87)	73 (61–83)	77 (19–88)	0.022

* Intubated patients at day 0 excluded. ** At 6 AM on the first day after ICU admission. *** Regardless of the type of oxygen support.

## Data Availability

Data available in agreement with steering committee PRoAcT–COVID-study.

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
