# Peer review of "Practice of Awake Prone Positioning in Critically Ill COVID-19 Patients—Insights from the PRoAcT–COVID Study"

_jcm, 2022, doi:10.3390/jcm11236988_

Round 1

Reviewer 1 Report (Previous Reviewer 2)

Dear Authors, 

Thank you for the amendments provided.

Author Response

Thank you for reviewing this manuscript.

Kind regards,

Willemke Stilma

Reviewer 2 Report (New Reviewer)

1. The rationale of the difference in treatment to the two patient groups should be provided (e.g. hospital policy, practice, decision to cardiovascular disease). Were patients having more severe disease or history decided for prone positioning? Does the history affect the outcome, e.g. probability of no treatment failure, even without prone positioning. COVID-19 can be a disease to complicate cardiovascular system (PMID: 33226078). Patients with underlying cardiovascular disease are more likely to develop severe COVID and poor outcome. This limitation should be discussed. How would this affect the conclusion for prone positioning should be discussed. 

2. It will be helpful to provide more scientific description to prone positioning. (e.g. how does it work and help to alleviate severe outcome).

3. For line 162 to line 188, it will be helpful to provide detail for how propensity score was calculated.

Author Response

Thank you for your reviewing our manuscript. A point to point reply is added. 

Kind regards, Willemke Stilma

Reviewer 3 Report (New Reviewer)

Thank you for the opportunity to review this paper that contributes knowledge about the practice of awake prone positioning in critically ill COVID-19 patients. I congratulate Authors with this well-designed study and transparency in reporting and commenting on study findings.

Minor suggestions follow:

·       The authors may consider mentioning PRoAcT - COVID already in the abstract.

·    Authors could further sustain the rationale to employ awake prone positioning in critically ill COVID-19 patients by adding data on the success of pronation in patients with severe COVID-19 early in the introduction.

·       Lines 90-91. Provide operational definition for acute hypoxemic respiratory failure.

·       Some information about SAPS scoring are needed

·       Who were the data collectors? What was the training like? Provide some information about the training methods as well as the length of the training.

·       Table 2 is redundant and can be deleted.

·       Table 3. Present the type of oxygen interface according to frequency criteria.

·       Lines 272-273 should be moved to the results section when describing the participating ICUs. Then, it is fine to comment how the wide range of ICUs contributes to the generalizability of the findings.

·       Line 296. Typo, amulticenter instead of a multicenter.

·       Line 299. Perhaps, is 5 a typo?

·       Reference 16 refers to ARDS; however, only some clinical features of COVID-19 are consistent with typical ARDS. I suggest looking for another reference consistent with respiratory failure due to COVID19.

Author Response

Tank you for reviewing our manuscript. A point tot point reply is added.

Kind regards, Willemke Stilma

Round 2

Reviewer 2 Report (New Reviewer)

The manuscript is improved. No further questions remain. 

This manuscript is a resubmission of an earlier submission. The following is a list of the peer review reports and author responses from that submission.

Round 1

Reviewer 1 Report

Your primary endpoint generally should not also be your intervention. If your two groups are prone-positioning and non-prone positioning then prone-positioning is your intervention and should not be your primary endpoint. If prone positioning is your primary endpoint are you doing any multi-variate analyses to determine what patients are more likely to reach this endpoint? Or merely reporting epidemiological data? 

Are subjects that are in the matched group as a non-prone subject but later experience proning added to the proning group as well? Or are these the 18 patients that are in the awake prone positioning sub-group but not the matched analysis?

If not the above explanation, can you explain the criteria that lead to 88 patients in you awake prone positioning group but only 70 in the matched analysis?

Could you please define intubated "immediately" after ICU admission. Is this 1 hour? 1 day? Similarly could you please define imminent transport? Given you are already excluding those transported on day 1 or 2, is this patients that were present for the full 2 days but left in the next hour? day?

How did you define awake prone positioning? Was it based on treatment orders or notes? Nursing documentation of patient position? If based on treatment orders or notes, do you control for patients who choose to sleep prone as their position of comfort? Is it known what percentage of adults in the Netherlands choose to sleep in a prone position? In the US it is 7-10% so if 10% of your "non-prone" group is actually spending 8ish hours a day prone that would be important to know.

In Table 4 you list Non-Rebreather or Venturi mask twice - at the top and the bottom. In addition your numbers of patients in the table only adds up to 87 and not 88.

In Table 4, your numbers of patients on none, Nasal sprong(sic), Non-rebreather or Venturi mask, HFNO and NIV do not add up to your totals. Overall only has 472, Prone only has 85 and Standard of Care only has 387. Your table legend and results section do not indicate why some subjects would be excluded. In addition, given the numbers add up to less than the total how did you classify patients who were on more than one type of support on the day of ICU admission? Is this the first support noted? The highest? The support at the end of the day?

Several of the studies that you cite that suggest a benefit to awake prone positioning define successful awake prone positioning as prone positioning in at least 2 consecutive days. However the median number of days of prone positioning and your entire IQR is only 1 day of prone positioning. I think this should be highlighted in your discussion. Did you consider doing a secondary analysis on the patients that had at least 2 days of prone positioning to match other studies?

Citation 1 does not mention hypoxemia, low-flow or high-flow oxygen supplementation. See line 52 and 53.

Citation 4 should not be included on line 60 as most of the patients were in intermediate or intensive care units and not on regular wards. 

Citation 9 specifically does include ICU patients. Either it should not be included on line 60 or the wording of the sentence on line 60 should be changed as the current wording of this sentence together with the following sentence suggests these studies were done only in the regular wards and not ICU.

Citation 10 does not specify if patients were in general wards or not so again, as above, questionable for line 60 unless wording is changed.

Your statement on line 60-62 that ICU data is scarce is questionable given the three references on line 60 all include ICU data and are larger than your study.

For citation 17 on line 269 you could add in many other citations to highlight the fact that many of the studies you cite average more than one day of awake proning.

Citations 10 and 19 are the same article

Author Response

Thank your for your valuable questions and suggestions. We have provided a point to point reply in the added file. 

Reviewer 2 Report

Dear Authors, 

Firstly, thank you for the opportunity to read this interesting paper. I acknowledge you for the idea of applying propensity matched analysis. I am convinced the issue of prone positioning is of utmost importance. However, conclusions must be drawn cautiously, especially in such restrospective analyses.

I believe your work would benefit from some changes and clarifications.

Firstly, it would be important to underline, that the most important data proving benefit of prone position in ARDS come from an RCT, in which prone position was performed for at least 16 hrs a day and the subjects received >4 sessions of proning. Moreover, the benefit was shown only in patients with P/F of <150 mmHg (in PROSEVA trial as well as in the post-hoc analyses of earlier studies from Gattinoni group). I believe this needs to be underlined very clearly in the discussion as both the studied populations and the intervention applied were very different from your study.

Furthermore, it would be very important to explain, what was the basis for the choice of position of the awake patients - dyspnea/hypoxemia/ROX/oxygen demand or just the decision of the clinician? Where there any local protocols or guidelines before the consensus you mention was reached? There must have been some principles in force as haemodynamics were clearly taken into account.

Moreover, I am not sure if I fully understand Table 4. SpO2 and PaO2 look as if the values were desribed exclusively in patients treated with NIV. Only after careful comparison with Table 3 you may conclude that these are values in the entire populations of PP vs non-PP subjects.

I think some data regarding respiratory drive would be of great importance. Do you have any data regarding RR, PaCO2 or ROX index?

Again, thank you for sharing this data. This is a very important piece of information. However, I strongly believe we should warn readers not to draw far-fetched conclusions.

Author Response

Thank you for the valuable questions and suggestions. A point to point reply has been provided in the attached file. 
